# Unconditionally teleported quantum gates between remote solid-state qubit registers

Mariagrazia Iuliano ⦿ , Nicolas Demetriou, H. Benjamin van Ommen ⦿ , Constantijn Karels ⦿ , Tim H. Taminiau ⦿ & Ronald Hanson ⦿ ✉

Quantum networks connecting quantum processing nodes via photonic links enable distributed and modular quantum computation. In this framework, quantum gates between remote qubits can be realized using quantum teleportation protocols. The essential requirements for such non-local gates are remote entanglement, local quantum logic within each processor, and classical communication between nodes to perform operations based on measurement outcomes. Here, we demonstrate an unconditional Controlled-NOT quantum gate between remote diamond-based qubit devices. The control and target qubits are Carbon-13 nuclear spins, while NV electron spins enable local logic, readout, and remote entanglement generation. We benchmark the system by creating a Greenberger-Horne-Zeilinger state, showing genuine 4-partite entanglement shared between nodes. Using deterministic logic, single-shot readout, and real-time feed-forward, we implement non-local gates without post-selection. These results demonstrate a key capability for solid-state quantum networks, enabling exploration of distributed quantum computing and testing of complex network protocols on full-stack systems.

Quantum networks can connect separate quantum processors to unlock capabilities and applications that do not have a classical counterpart. Examples range from long-range secure communication and distributed quantum computing to enhanced quantum sensing[1,2]. In particular, distributed quantum computing exploits quantum links between small quantum processors to build larger networks that allow the system to scale in size or distance[3,4]. Key to such modular architectures are non-local quantum operations, which can be performed using quantum teleportation protocols[5,6]. Quantum gate teleportation (QGT) poses stringent requirements on the qubit platform, including distribution of remote entanglement, executing local operations within a multi-qubit register and performing non-local feed-forwarded operations within the coherence time of its qubit register. To avoid low gate success probabilities and ensure scalability, QGT should run unconditionally on the outcomes of the mid-circuit measurements of the teleportation protocol. This implies that once entanglement is shared between the processors, the gates should operate deterministically.

Pioneering experiments have demonstrated probabilistic remote QGT in purely photonic systems[7,8] as well as with photonic systems combined with quantum memories[9-11]. These demonstrations are readily extensible to longer distances, but could not achieve unconditional operation as they inherently rely on post-selection. Unconditional (and even fully deterministic) QGT has recently been achieved within a single cryogenic system with superconducting qubits[12,13], within a segmented ion trap system[14] and between nearby trapped ion systems[15].

Here, we implement unconditional QGT between solid-state qubits across an extensible optical link. In particular, we employ Nitrogen-Vacancy (NV) spin qubits in diamond. This platform has previously enabled heralded entanglement generation over 10 km distance using 25 km of deployed fiber[16], as well as the realization of basic network protocols on a three-node network[17,18].

We first generate a 4-qubit Greenberger-Horne-Zeilinger state using two independently controlled two-qubit registers. Each register consists of an NV center electron spin qubit and a $^{13}C$ nuclear spin qubit, housed in separate cryostats (Fig. 1a). We then perform the

QuTech & Kavli Institute of Nanoscience, Delft University of Technology, Delft, the Netherlands. ✉e-mail: r.hanson@tudelft.nl

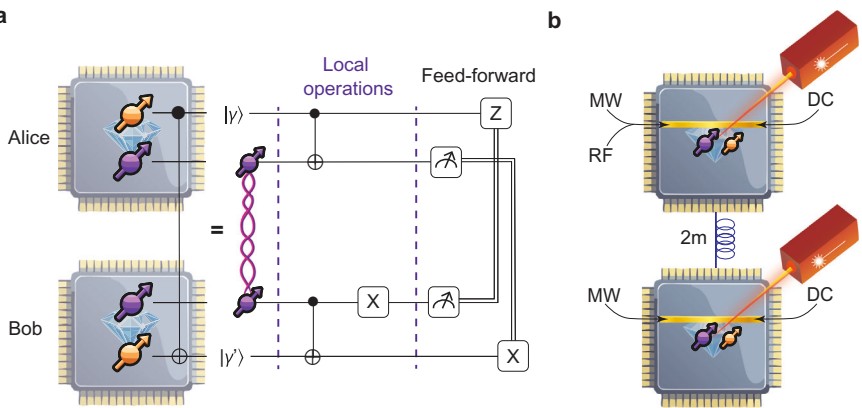

**Fig. 1 | Experiment and resources overview. a** C-NOT quantum gate teleportation: we use two separated nodes based on NV-center defects in diamond. Each node is composed of two qubits: one communication qubit (in purple), obtained via controlling the electron spin of the NV, and one data qubit (in yellow), made by controlling a single $^{13}$C nuclear spin. To realize a non-local C-NOT gate between the data qubits, a teleportation protocol is used, including the generation of remote entanglement, local operations and feed-forward operations. **b** Concept of the

control setup for the two-qubit register on each node, separated by 2m of optical fibers. The qubits are manipulated via microwaves (MW) in GHz range, and additionally radio-frequency (RF) waves for Alice's data qubit (in MHz range), sent along a gold stripline. Preparation, readout and entanglement generation require optical control via red (637nm) and yellow (575nm) lasers, whose outputs are combined in a single excitation optical path. A DC voltage is applied to use the Stark effect for tuning the emitted photon frequency of the two nodes.

teleportation of a Controlled-NOT quantum gate between the two remote nuclear spin qubits. In both cases, we exclude post-selection and data filtering, unconditionally accepting all intermediate measurement outcomes, and use real-time feedforward operations within the registers' coherence time. This demonstration of unconditional QGT is made possible by several innovations compared to previous NV center network experiments[17–19], including tuning of the optical transition frequency at high-magnetic-field, different tailored control methods for the nuclear spin qubits in the two nodes: dynamical decoupling (DD)[20] and dynamical decoupling-radio frequency[21,22] (DDRF), in combination with remote entanglement generation and node synchronization, and improved nuclear spin qubit phase tracking strategies (see Sec. 2.3) during network activity.

## Results

We employ two setups (Alice and Bob) hosting diamond NV centers that are physically separated by 2 m of optical fiber in a lab and cooled down to $T_{Alice}$= 3.9 K, $T_{Bob}$= 3.4 K (Fig. 1b). The electron spin qubits, referred to hereafter as communication qubits, are manipulated using microwave (MW) pulses delivered on-chip via gold striplines. Initialization and single-shot readout of these qubits are performed via spin-selective optical transitions[23].

### Nuclear spin control

In addition to the communication qubit, each node employs a hyperfine-coupled $^{13}$C nuclear spin as a data qubit. The Hamiltonian that describes the interaction between the electron spin qubit and the nuclear spin qubit is approximated by[20]:

$$H = \omega_L I_z + A_\parallel S_z I_z + A_\perp S_z I_x \qquad (1)$$

where $\omega_L = \gamma B_z$ is the Larmor frequency of the nuclear spin in the external magnetic field $B_z$. The external magnetic field for Alice (Bob) is 189 mT (31 mT). $S_i$ and $I_i$ are the spin operators for the electron spin and the nuclear spin, respectively. $A_\parallel$ and $A_\perp$ are the parallel and perpendicular hyperfine coupling parameters (more details in the Supplementary Note 2).

We optimize the control of the data qubits by using two different techniques. At Alice, we use the DDRF method[21,22], in which the data qubit is directly driven via phase-controlled RF pulses, interleaved with Dynamical Decoupling sequences to protect the communication qubit from decoherence. Bob's data qubit, instead, is manipulated using

tailored DD sequences, therefore achieving control via the communication qubit dynamics[20,24,25]. For Alice, the high magnetic field regime provides significant advantages in qubit control. In this regime, the DDRF technique enables control of nuclear spins with small $A_\perp$ (compared to $\omega_L$). The DDRF gates bring versatility and multi-qubit control while showing similar gate fidelity as the DD gates used on this qubit in ref. 18. Here, we exploit the feature that the gate duration is easily adaptable to timing constraints set by the other node, contributing to optimized experimental rates and higher overall system fidelity. Additionally, when DDRF is combined with remote entanglement generation (see Sec. 2.3 and Supplementary Note 3), this enables less complex and more efficient phase tracking of the data qubit.

In Fig. 2a and b, we show the gate sequences to initialize and read out the data qubits. The sequences are control technique-independent, unless otherwise specified. Both the initialization and read-out sequences are assisted by the communication qubit. The Z-gates on the data qubit for Alice are performed by updating the phase on the local oscillator of the RF field, and for Bob by either waiting a certain amount of time or playing a specific DD sequence based on the phase we want to imprint. In Fig. 2c, we show the measured fidelity of each data qubit with the ideal state for initialization in six unbiased states along the Bloch sphere: $\pm Z$, $\pm X$, $\pm Y$. We achieve average fidelity, corrected for known tomography errors on the communication spin[17], of 85(1)% for Alice and 96(1)% for Bob.

The main sources of infidelity are pulse errors on the communication qubit, leakage of laser light causing communication qubit dephasing, errors in the mapping of the state of the data qubit onto the communication qubit, and the imperfect decoupling of the communication qubit from the surrounding nuclear spin bath.

### Remote entanglement generation

For the generation of remote entanglement, the emission of indistinguishable photons from the remote communication qubits is critical. We introduce DC Stark tuning[26] on both setups to achieve indistinguishability in photon frequency, together with charge repumping using 575nm light on resonance with the Zero-Phonon Line of the neutral charge state (NV$^0$) to counteract ionization. The novelty of DC Stark tuning at high magnetic field is enabled by efficient charge repumping using a high-power 575nm pulse (see Supplementary Note 1) together with operating in favorable strain conditions.

Remote entanglement between the two nodes is generated using photonic number-state encoding[27–29]. The experimental

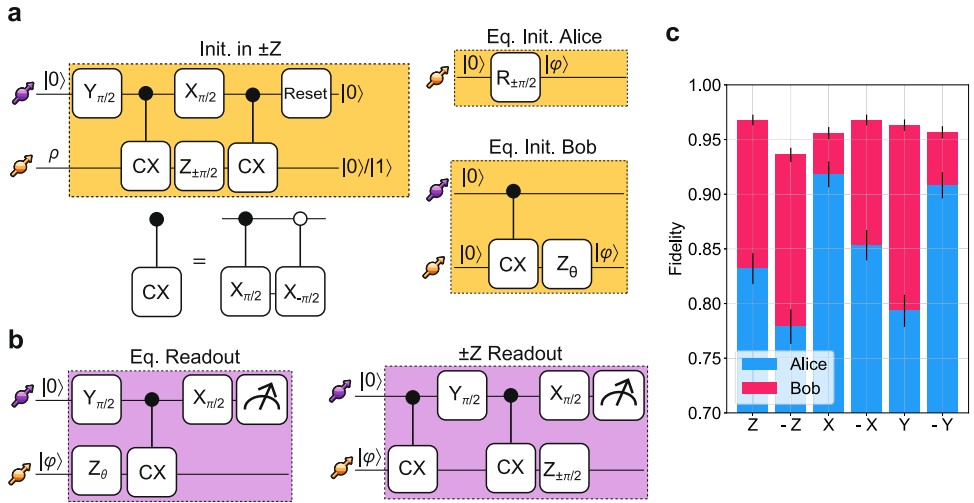

**Fig. 2 | Data qubit preparation. a** Data qubit initialization sequence. $\pm Z$ initialization with the electron spin qubit in $|0\rangle$ deterministically enables the initialization in one of the two eigenstates. The initialization gate is completed when the electron spin qubit is optically reset to the state $|0\rangle$. Initialization of any state $|\phi\rangle$ on the equatorial plane is obtained by adding an unconditional gate for Alice along a tailored combination of $\hat{x}$ and $\hat{y}$ axes when initialized in $|0\rangle$, or using a conditional gate and a phase gate with an arbitrary angle $\theta$ for Bob. **b** Readout of the data qubit. The state of the data qubit is mapped on the communication qubit and then optically read out. **c** Measured fidelity with the ideal state for a set of unbiased initial states along the Bloch sphere. Error bars represent one standard deviation.

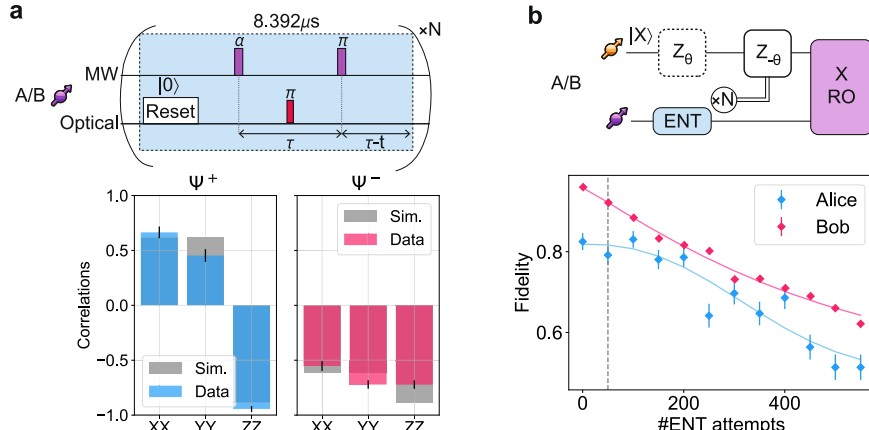

**Fig. 3 | Network activity characterization. a** Remote entanglement generation and entangled state fidelity. At each node, a single attempt includes a reset pulse to initialize the communication qubit in $|0\rangle$, a MW $\alpha$-pulse, which brings the qubit in an unbalanced superposition state; a short (1 ns) optical $\pi$-pulse that excites the population in the $|0\rangle$ state to the excited state, enabling spontaneous emission of a single photon; a MW $\pi$-pulse played at a time $\tau$ after the $\alpha$-pulse and $\tau$ before the next reset pulse in the subsequent attempt, hence a distance $\tau - t$ from the end of a single attempt, with $t$ indicating the time necessary to reset the electron spin state. The total duration of a single attempt is 8.392 $\mu s$ (details in the Methods section), which is repeated $N$ times. Lower panel shows measured and simulated correlations. **b** Characterization of the nuclear spin dephasing during entanglement attempts. During each entanglement attempt, the nuclear spin gains a deterministic phase, which we correct based on the number of repetitions $N$ before entanglement is heralded. Additional stochastic phases, e.g. due to the spin reset, cause decoherence. The plot shows the state fidelity of the nuclear spin state, initialized in a superposition state, for different numbers of entanglement attempts. The dashed grey line represents the chosen timeout for entanglement generation $N_{max}$=50. Error bars represent one standard deviation.

sequence, depicted in Fig. 3a, involves the generation of electron spin-photon entangled states at each node, in the form of $\sqrt{\alpha}|0\rangle_c|1\rangle_p + \sqrt{1-\alpha}|1\rangle_c|0\rangle_p$, where $|i\rangle_c$ and $|i\rangle_p$ are the communication qubit and photonic qubit states, respectively, and $\alpha$ is a parameter set in experiment. The spontaneously emitted photons travel towards a mid-point station, composed of a 50:50 in-fiber beam-splitter, whose output ports are connected to Superconducting Nanowire Single-Photon Detectors (SNSPDs). The detection of a single photon heralds, in a perfect scenario, the two-qubit state $(|01\rangle_c \pm e^{i\phi}|10\rangle_c)/\sqrt{2}$, with probability (and hence state fidelity) of 1-$\alpha$. Here $\phi$ is the optical phase difference between the two paths at the beam splitter, which is actively stabilized before entanglement generation[17]. The sign of the entangled state depends on which detector clicked.

In Fig. 3a we report the measured values of the entangled state correlators along with their simulated values for the states $\Psi^+$ and $\Psi^-$. We obtain state fidelities of 77(2)% and 76(2)% for $\Psi^+$ and $\Psi^-$ respectively, with an average $\alpha$=0.045 between the two setups. For comparison, the average simulated state fidelity is 79%. Detailed explanations about the protocol, the source of errors and the simulated values are discussed in ref. 29.

### Data qubit coherence during networking
The data qubits, encoded in nuclear spins, possess a long intrinsic coherence time (tens of milliseconds for the current devices). However, during entanglement attempts, the coherence of the data qubit undergoes a faster decay due to its coupling to the communication qubit whose state cannot be perfectly tracked in entanglement

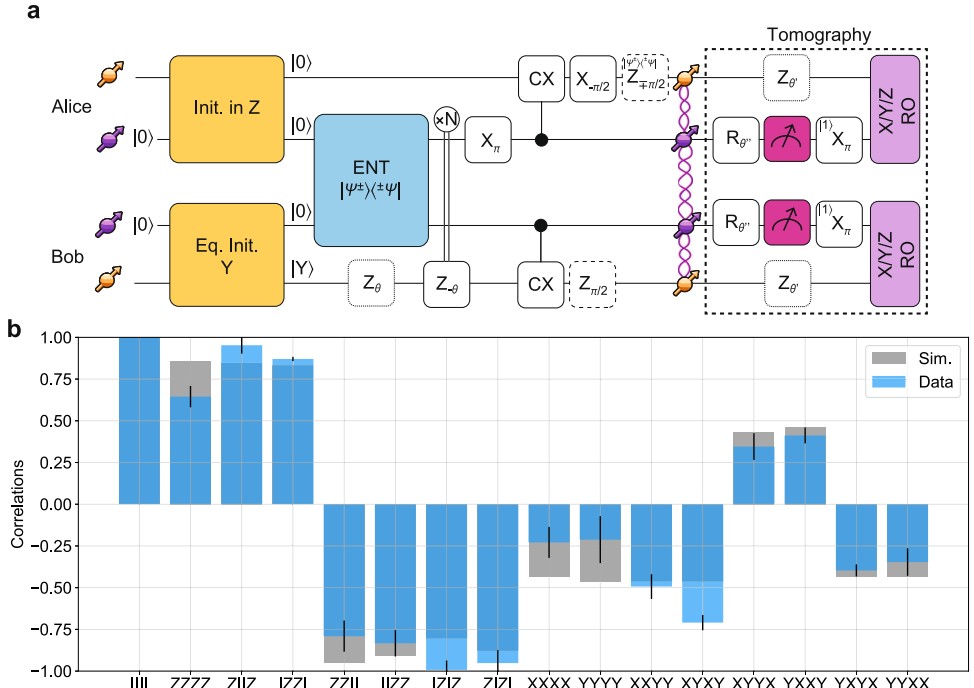

**Fig. 4 | Realization of a remote 4-qubit GHZ state. a** Circuit diagram. The data qubits are initialized using the sequence in Fig. 2a. After heralding entanglement, a rephase gate is played on Bob's data qubit. Subsequently, a set of local operations completes the generation of the GHZ state. Dashed gates represent gates that are not individually executed, but are compiled in the readout sequence. To measure the correlators in (**b**), we first measure the electron spin state, using single-qubit gates for the measurement basis selection and a non-destructive optical readout (highlighted in magenta). Every outcome is accepted. If the outcome is $|1\rangle$, a $\pi$-pulse flips the state to ensure the assisted-readout always starts with the communication qubit in $|0\rangle$. During the readout of the electron spin qubit, the data qubit picks up another phase $\theta'$ depending on the measurement outcome, whose rephasing is also compiled in the subsequent assisted-readout. **b** GHZ correlator results and corresponding simulated values. Error bars represent one standard deviation.

attempts[30]. The dephasing time under network activity is parametrized by the number of entanglement attempts $N_{1/e}$ after which the fidelity contrast of the state stored in the data qubit has decreased by $1/e$. During an entanglement attempt, the time that the communication qubit is in $|0\rangle$ versus $|1\rangle$ is not deterministic, decomposing the total phase acquired by the data qubit in a static offset plus stochastic variations. Therefore, real-time tracking of the phase becomes critical (Fig. 3b) and $N_{1/e}$ is thus affected by the accuracy of the nuclear spin evolution phase tracking.

For Alice, the phase tracking is executed on the local oscillator of the data qubit RF driving field, updating the phase of the next RF pulse. The average phase picked up during a single entanglement attempt is calibrated beforehand. For Bob's data qubit, the rephasing after entanglement attempts is achieved via an XY8 DD sequence on the electron spin, in which the inter-pulse delay is tailored to result in the specific phase we want to imprint on the nuclear spin evolution[18]. Additionally, it is important to protect the communication qubit during this process and therefore it is key to avoid inter-pulse delays for which the communication qubit couples to other nuclear spins in its environment. The optimized inter-pulse delays are also calibrated beforehand and compiled in a look-up table for the control device (details in Supplementary Note 3).

In Fig. 3b we report the fidelity of the input state on the data qubit as a function of the number of entanglement attempts while employing the above-mentioned rephasing techniques. We extract the parameter $N_{1/e}$ by fitting the data to the exponential decay curve $A \cdot e^{-(n/N_{1/e})^d} + 0.5$, where $A$ is related to the initial fidelity and $d$ is the exponential decay. We obtain a $N_{1/e}$ of 391(31) (479(19)) for Alice (Bob) with $d$ of 2.4(7) (1.1(1)) and $A$ equals 0.32(2) (0.46(1)). Based on these results, we set the timeout for the entanglement generation to 50 attempts before re-initializing the data qubit. The choice of the time-out is a trade-off between the experiment rates and corresponding

fidelities. We note that the coherence time during entanglement attempts may be further prolonged by introducing dynamical decoupling pulses for the data qubit, as shown in refs. 18,19.

## 4-qubit GHZ state

Next, we combine all the above techniques for the creation of a 4-qubit GHZ state distributed over 2 nodes. Besides demonstrating the generation of a crucial resource state for quantum information protocols[31], this experiment serves as a system benchmark for the non-local C-NOT gate, as it utilizes the same gate set for local operations, together with fixed sequences for initialization, remote entanglement generation, rephasing of the data qubit after entanglement using real-time feedforward, mid-circuit readout of the communication qubit and data qubit readout.

The circuit diagram in Fig. 4a shows the gate sequence for the creation of the state $\Psi_{GHZ} = 1/\sqrt{2}(|0\rangle_{Ad}|1\rangle_{Ac}|1\rangle_{Bc}|0\rangle_{Bd} - |1\rangle_{Ad}|0\rangle_{Ac}|0\rangle_{Bc}|1\rangle_{Bd})$, with $A$ ($B$) indicating the node Alice (Bob) and $c$ ($d$) the communication (data) qubit in each node. The initialization of the data qubit is achieved via the circuits shown in Fig. 2a. To ensure that both nodes enter the remote entanglement generation sequence at the same time, the initialization of the two data qubits is synchronized by delaying the start of the initialization of the fastest node. After successful entanglement generation, Bob's data qubit is rephased based on the number of entanglement attempts used. In case the generated remote entangled state is $\Psi^-$, the midpoint communicates this to Alice where an extra phase gate is added in real time to the tomography pulses of the data qubit. Effectively, this ensures that the remote entangled state is $\Psi^+$ irrespective of the photon detection pattern. Next, $\Psi^+$ is transformed into $\Phi^+$ by a Pauli correction gate applied at Alice. Subsequently, local operations on the qubit registers are performed that entangle the data qubits with the communication qubits. Phase gates on the data qubits at the end of the protocol are compiled

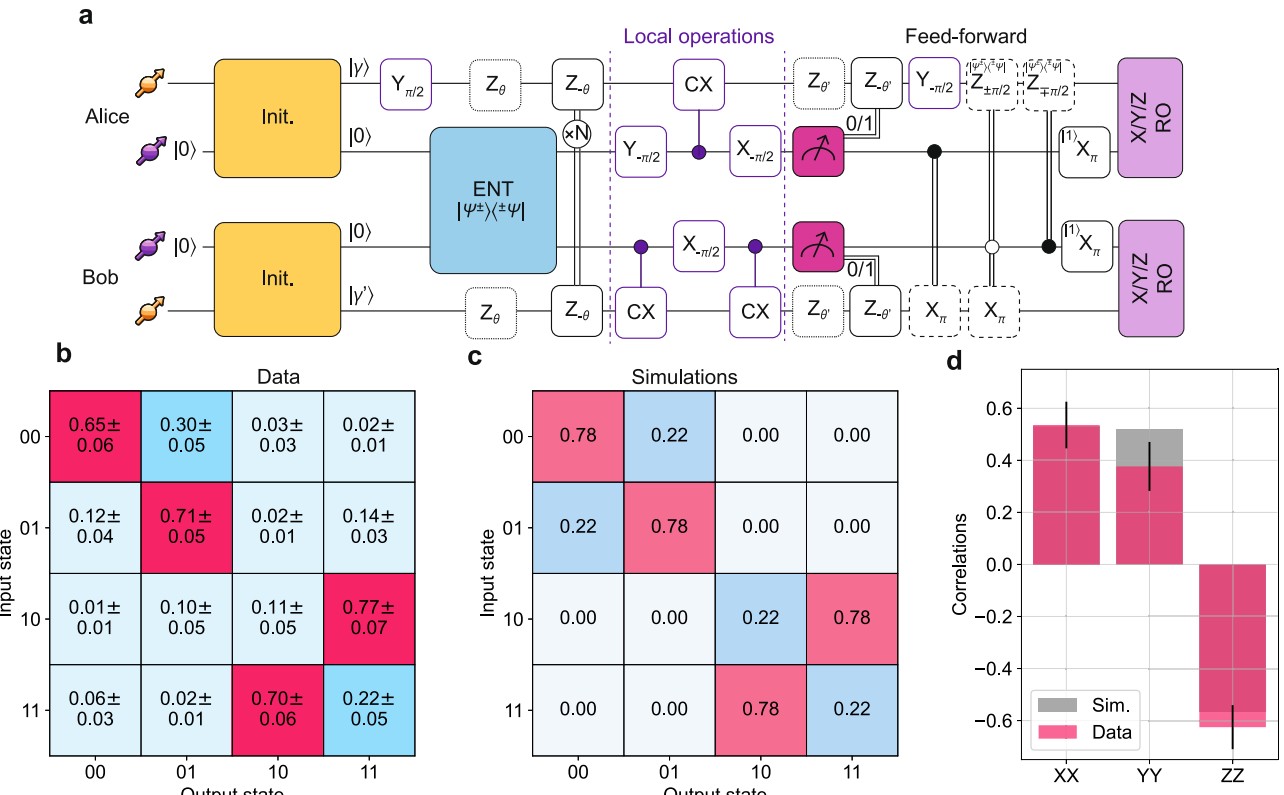

**Fig. 5 | Non-local C-NOT gate. a** Circuit diagram using native NV gates. The gates in purple compile a local C-NOT gate. For Alice, the first (last) unconditional gate for the local operations is executed before the entanglement generation (after the mid-circuit readout) for synchronization purposes. The dotted gates represent the phase acquired by the data qubit during entanglement or during the readout of the communication qubits, and are followed by the respective rephasing gates. The feed-forward operation (dashed gates) is compiled in the readout sequence. The magenta mid-circuit measurement indicates a non-destructive readout. **b** Measured classical truth-table. The initial states on the data qubit are the eigenstates and we report the non-local two-qubit state fidelity. As expected, we see a bit-flip in Bob's state when Alice's input state is $|1\rangle$. **c** Simulated classical truth table. **d** Generation of an entangled state via the non-local C-NOT gate. We prepare the data qubits in $|X\rangle_A$ and $|1\rangle_B$, to obtain the entangled state $\Psi^+$. The histogram shows the correlator expectation values together with their simulated values. Error bars represent one standard deviation.

into the final tomography pulses. Experimental details of the tomography are discussed in the Methods section.

In Fig. 4b we report the measurement results of the 4-qubit correlators, along with the predicted values from simulations using measured parameters. This data as well as data presented below is corrected for known tomography errors (see Supplementary Notes 4 and 5 for details). We obtain a state fidelity $F_{GHZ}=64(4)\%$, in good agreement with the value predicted from simulations of $F_{GHZ}^{sim}=66\%$. The observed value of $F_{GHZ}$ exceeding 0.5 proves the generation of genuine four-partite entanglement across the two nodes[32]. We emphasize that this state is generated without any post-selection, constituting to the best of our knowledge the largest heralded GHZ state across optically connected solid-state network nodes demonstrated so far.

The GHZ state fidelity is mainly limited by imperfections in the remote entangled state generation and initialization of the data qubits. Separately, incorrect state assignment of the communication qubit measurement outcome in the tomography leads to a wrong rephasing sequence applied to the data qubit. We estimate that this occurs for ~5% (~9%) of the measured $|1\rangle$ outcomes for Alice (Bob), causing tomography errors that reduce the observed state fidelity by ~7%. We thus estimate that the actual GHZ state fidelity is about 71%.

**C-NOT gate teleportation**

We realize a C-NOT gate between the data qubits of the two remote nodes, using the gate circuit shown in Fig. 1a. Compared to the GHZ state generation, we add real-time feed-forwarded operations

based on the exchange of classical information between the nodes. In Fig. 5a, we report the circuit diagram presented in Fig. 1a translated into native gates of our platform. Note that of the local operations (gates depicted with a purple boundary), the single-qubit gates on Alice's data qubit are executed right after the initialization and right after the mid-circuit measurement. This compilation optimizes the synchronization between the nodes taking into account the different gate durations on the two nodes. This synchronization is required not only during the entanglement attempts (as in the GHZ case) but also when exchanging classical information for the real-time feed-forward operations.

We first reconstruct the classical truth table of the C-NOT gate. For this, the initial states prepared on each data qubit are the two eigenstates $|0\rangle$ and $|1\rangle$. On Bob's side, this results in the qubit not being subjected to additional dephasing during the entanglement attempts. In contrast, on Alice's side the data qubit is in a superposition state during the network activity, due to the local gate being executed before the entanglement generation as discussed above; therefore, the dephasing mechanisms and the phase tracking reported in Fig. 3b are relevant.

The results of the truth table measurements are displayed in Fig. 5b. For comparison, we include in Fig. 5c the simulated truth table. The results show the correct gate action with the four two-qubit fidelities being above 70% on average, in reasonable quantitative agreement with the simulations.

Subsequently, we show the quantum-coherent nature of the non-local C-NOT gate by generating an entangled state between the data qubits. Specifically, we prepare Alice's data qubit in $|X\rangle$ and Bob's data

qubit in $|1\rangle$. Application of the non-local C-NOT generates the two-qubit entangled state $\Psi^+$ in the ideal case. We analyze the resulting state by measuring the two-qubit correlators $\langle XX \rangle$, $\langle YY \rangle$ and $\langle ZZ \rangle$. The experimental results are shown in Fig. 5d, together with the simulated values. We then extract the state fidelity $F_{\Psi^+} = (1 + \langle XX \rangle + \langle YY \rangle - \langle ZZ \rangle)/4$, where $\langle ii \rangle$ represents the measured correlator. We find a state fidelity $F_{\Psi^+} = 63(4)\%$, in good agreement with its simulated value of $F_{sim} = 65\%$, demonstrating entanglement between the remote data qubits.

The main sources of error for the experiments in this section are the same as in the GHZ state generation. In addition, wrong assignment of the mid-circuit readout results in a wrong feed-forward operation on the data qubit and an error to the gate. To quantify the corresponding infidelity, we simulate the scenario of accepting only $|00\rangle_c$ mid-circuit readout results. We find that, in this case, the expected average fidelity for the classical truth table outcomes is 90%, while the expected entangled state fidelity reaches 76%[33], indicating that an improved readout would yield significant gains in gate performance.

## Discussion and outlook

This work demonstrates the realization of heralded genuine four-partite entanglement and the implementation of an unconditionally teleported quantum gate, adding key capabilities for solid-state quantum network testbeds that open up several new avenues. Taking the current platform as a basis, the number of data qubits per node can be further increased. In particular, the DDRF control method, integrated here with a network link, enables extension to multi-qubit control[21], enabling the generation of larger resource states that could be used, for instance, for exploring error correction on a distributed processor[34].

Another interesting direction is towards fully deterministic non-local gate operation, without imposing a timeout on the entanglement generation attempts and re-initializing the data qubits when the entanglement generation does not succeed within the timeout. This requires an active link efficiency exceeding one[35], meaning that the data qubit coherence time under network activity has to exceed the time required to generate one (or more) entangled states. The active link efficiency can be improved both by extending the data qubit coherence and by enhancing the remote entanglement generation rate. For the former, recent experiments on a weakly coupled $^{13}$C nuclear spin[35] as well as on a data qubit encoded in a pair of nearby $^{13}$C nuclear spins[36] promise orders of magnitude improvement in coherence under networking activity. Integrating such data qubits into non-local protocols directly benefits from the phase tracking developed here. For entanglement generation, both cavity enhancement[37,38] and employing more efficient communication qubits[39–44] can lead to substantial rate enhancements. The techniques and methods developed in the current work can aid and accelerate the development of other communication qubits, such as the DDRF techniques pioneered on the current platform being adopted to diamond group-IV qubits[45].

Following earlier integration tests of this platform with software control layers[46,47], the current work also impacts quantum network stack development. Both the 4-qubit GHZ resource state and the non-local gate operations expand the set of network protocols that can be explored and tested using higher layers of the stack. Scaling the number of available qubits and enabling more complex applications also opens the way to experimentally investigate optimal network synchronization and classical communication strategies, as well as network application compilers[48–50].

## Methods
### Experimental setup
The setup utilized in this work is based on the setup of Bob and Charlie nodes in refs. 17,18. More details are included in the Supplementary Note 1.

### Remote entanglement generation duration
As shown in Fig. 3a, the duration of a single remote entanglement attempt is 8.392 $\mu$s. The length of a single entanglement attempt $L$ is set by the required decoupling time $\tau$, the duration of the reset pulse $t_{reset}$, and the time $t$ necessary to reset the electron spin state. $t_{reset}$ includes the actual on-time of the laser field and the response time of the acousto-optical modulator to make sure that the reset pulse is completely off when the first microwave pulse is applied. This constrains $L = 2\tau + t_{reset} - t$ and $L$ must be the same for both nodes. Consequently, a free parameter for each node is $\tau$. Given that Bob experiences a lower magnetic field compared to Alice, its minimum $\tau$ is ~ 3.0 $\mu$s, which effectively sets the minimum allowed duration as $L \geq 2\tau$. Additionally, $\tau$ must be chosen to avoid undesired coupling to surrounding nuclear spins. As a result, for Alice the value of $\tau$ is adapted to fulfill the duration $L$ set by Bob.

### Qubit readout
The tomography basis-selection on the electron spin is executed via a single MW pulse with axis and angle depending on the chosen readout basis, while the optical readout is performed using long weak laser pulses ( ~ 0.1 nW for up to 190 $\mu$s) with a dynamical stop on the laser field when a single photon is detected. This method ensures a non-destructive readout, crucial for avoiding additional dephasing on the data qubit. Both outcomes, $|0\rangle$ and $|1\rangle$, are accepted, but for outcome $|1\rangle$, the communication qubit is afterward flipped to ensure it is always in the $|0\rangle$ state for the assisted-readout of the data qubit. During the readout of the communication qubit, the data qubits are picking up a phase depending on the outcome of the readout and its duration. This phase is also compiled in the communication qubit-assisted readout for the data qubit tomography. For the final readout on the communication qubit, after mapping the state of the data qubit on it, we use a shorter and higher power pulse ( ~ 1nW for up to 40 $\mu$s), without dynamical stop.

### GHZ state fidelity
The 4-qubit GHZ state fidelity, provided that the measured correlators $C_i$ signs are in accordance with the expected state, is calculated as:

$$F_{GHZ} = \frac{\sum_{i=1}^{16} |C_i|}{16} \qquad (2)$$

while the error is propagated as:

$$\sigma_{GHZ} = \left( \sqrt{\sum_{i=1}^{16} \sigma_{C_i}^2 + 2 \cdot \sum_{\substack{i=2}}^{8} \sum_{\substack{j=2 \\ j \neq i}}^{8} Cov(C_i, C_j)} \right) / 16 \qquad (3)$$

where the covariance term takes into account the full correlations among the Z terms, as they are directly extracted from the $\langle ZZZZ \rangle$ measurement.

## Data availability
The data generated in this study have been deposited in the 4TU.ResearchData database.

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

## Acknowledgements

The authors thank K.L. van der Enden, J. Fischer, S.L.N. Hermans, A.R.-P. Montblanch, A.J. Stolk, C. Waas, J. Yun for experimental support and fruitful discussions. The authors gratefully acknowledge support for the research of this work from the joint research program "Modular quantum computers" by Fujitsu Limited and Delft University of Technology co-funded by the Netherlands Enterprise Agency under project number PPS2007, from the Dutch Research Council (NWO) through the Spinoza prize 2019 (project number SPI 63-264), from the European Union's Horizon Europe research and innovation programme under grant agreement No. 101102140 - QIA Phase 1 and from the European Research Council (ERC) under the European Union's Horizon 2020 research and innovation programme (grant agreement No. 852410).

## Author contributions

M.I. and R.H. devised the experiments. M.I. prepared the setup (hardware and software), carried out the experiments, collected and analyzed data. Analysis and results were discussed between all authors. M.I. developed the DD control for this experiment with input from N.D. N.D. developed the DDRF control with input from M.I. and H.B.v.O. M.I. and C.K. developed the Stark tuning control at high magnetic field. M.I. and R.H. wrote the main manuscript with input from all the authors, M.I. wrote the Supplementary Material with input from all the authors. T.H.T. and R.H. supervised the research.

## Competing interests

The authors M.I., N.D., H.B.v.O., T.H.T. and R.H. are listed as inventors for the patent WO2025206952A concerning the DDRF phase tracking method utilized in this work. R.H. is co-founder and shareholder of Delft Networks B.V.

## Additional information

**Peer review information** : *Nature Communications* thanks Hideo Kosaka, and the other, anonymous, reviewer(s) for their contribution to the peer review of this work. A peer review file is available.

