## [Transparent Peer Review file · Nature Communications]

Unconditionally teleported quantum gates between remote solid-state qubit registers

Corresponding Author: Professor Ronald Hanson

Version 0:

Reviewer comments:

Reviewer #1

(Remarks to the Author)

This manuscript reports a milestone in distributed quantum computing by demonstrating an unconditional, non-local CNOT gate between two remote diamond-based nodes. Unlike previous post-selected operations, the authors achieve a deterministic protocol by integrating high-fidelity remote entanglement, single-shot ^{13}C nuclear spin readout, and real-time feed-forward control. The gate's robustness is verified through the creation of a four-partite GHZ state, confirming genuine entanglement across the remote registers.

The study's primary contribution lies in its real-time phase tracking and feedback, which mitigates dephasing during probabilistic entanglement generation. By enabling complex multi-node operations without post-selection, this work provides a scalable framework for error-corrected distributed quantum computation and the development of a functional quantum internet.

This manuscript addresses a topic of significant contemporary interest within the quantum information science community, and I anticipate it will attract substantial attention from researchers across the field. The authors present a study that is both technically sound and methodologically rigorous; the experimental results are solid and provide convincing evidence for their claims. Given the high quality of the work and the clarity of the presentation, I find the results to be robust and have no further technical concerns or requests for revisions.

However, before the publication, I have questions.

(1) In the implemented real-time phase tracking, for how long can the cumulative tracking error be tolerated? Furthermore, is this precision maintained in cases requiring even longer communication attempts, such as when the distance between nodes is further increased?

(2) I have a particular interest in the scalability of the proposed system. While the current experiment successfully utilizes two NV-center nodes, could the authors comment on how the inherent inhomogeneities between nodes—specifically variations in optical transition wavelengths and hyperfine coupling strengths to nuclear spins—might impact the overall gate performance? Furthermore, do the authors anticipate that such inhomogeneities will present significant challenges when scaling the network to three or more nodes? If so, I would appreciate the authors' insights on potential strategies or compensation mechanisms to mitigate these effects in future large-scale implementations.

(Remarks on code availability)

Reviewer #2

(Remarks to the Author)

The manuscript reports an impressive experimental advance toward distributed solid-state quantum computing, namely the realization of an unconditional teleported non-local C-NOT gate between remote diamond-based qubit registers, together with the generation of remote four-qubit entanglement. The work is timely, technically strong, and of clear interest to the quantum networking and modular quantum computing communities. I have only a few minor comments, mainly concerning

figure clarity and consistency of notation/presentation, which I believe can further improve the readability of the manuscript.

Figures 1 and 4a (local operations).

In Figs. 1 and 4a, the assignment/order of the local-operation blocks between Alice and Bob seems visually inconsistent, or at least unclear. Please check whether the upper and lower local-operation parts are swapped, or clarify the control/target convention more explicitly in the figure and/or caption.

Figure 2 (CX_($\pi/2$) notation).

In Fig. 2, the notation CX_($\pi/2$) is not immediately clear. If this is meant to denote an ordinary controlled-X gate in the native decomposition, using simply CX would likely improve readability. Otherwise, the meaning of the $\pi/2$ subscript should be explicitly defined in the figure caption or main text.

Figure 2a,c (Z_θ versus Z_φ).

Please check the consistency of the phase-angle notation in Fig. 2, in particular Z_θ versus Z_φ. The use of θ may be intentional, but the notation should be made consistent across the manuscript to avoid confusion.

Figure 3a (undefined t).

In Fig. 3a, the symbol t appearing in the expression τ-t does not seem to be defined in the main text or figure caption. Please define it explicitly in the caption or main text, even if it is explained in the Supplementary Information.

Figure 3b (vertical axis p_0).

In Fig. 3b, the vertical axis is labeled p_0, whereas both the caption and the main text appear to describe the plotted quantity as the state fidelity. Please make these consistent and clarify whether the figure shows the $|0\rangle$ population or the state fidelity.

Figure 5 / main-text description of Alice's single-qubit gates.

The description of the timing of the single-qubit gates on Alice's data qubit appears ambiguous. The main text states that these gates are executed "right after the initialization and right after the mid-circuit measurement," whereas the circuit seems to show more than two such gates, and the final Y gate appears to occur after the feed-forward sequence rather than immediately after the measurement itself. Please clarify the timing and compilation of these gates in the figure and/or text.

Overall, these are relatively minor issues, and I believe the manuscript will be further strengthened once the above points are clarified. The reported results are significant and will be of broad interest to readers working on quantum networks, distributed quantum processing, and solid-state quantum platforms.

(Remarks on code availability)

Reviewer #3

(Remarks to the Author)

The authors realise a Controlled NOT gate between two remote nuclear spin qubits in two separate experimental setups, a crucial step toward scalable modular quantum processors. They also produced a four-partite GHZ state across two nodes, which is the largest heralded multi-partite entangled state for solid-state systems to date. The ability to operate without post-selection and using real-time feedback enable the realization of this key milestone for distributed quantum computing. The experiment is highly advanced and requires sophisticated protocols and demanding control. Overall, this is a highly relevant and significant work. The main text is somewhat hard to read, originating from the many advanced techniques involved which are not explained in detail.

I am convinced this is a valuable contribution to Nature Communications, and I recommend publication.

I only have minor comments:

- Abstract: What should fully-integrated system mean? This doesn't seem to be a particular property of the reported experiment.
- Use of capital letters (RadioFrequency, MicroWave, Dynamical Decoupling) is somewhat unusual
- Power-broadened 575nm pulse – unclear wording – probably the pulse power-broadens the transition?
- How is DC Stark tuning at high mag field enabled by efficient charge repumping? Unclear sentence, please rephrase / explain
- The scheme requires classical communication – how limiting will this be for long-distance links?
- What limits the fidelity of the data qubit initialization? It is rather low for Alice (85%), and a brief discussion would be helpful in the main text.
- What limits the fidelity of the entanglement generation? At least a brief statement would be helpful in the main text.
- What is the entanglement generation rate? Please give a value
- What is the success probability per attempt of successful remote CNOT gates?

(Remarks on code availability)

it contains a readme

Response to referees:

Reviewer #1 (Remarks to the Author):

This manuscript reports a milestone in distributed quantum computing by demonstrating an unconditional, non-local CNOT gate between two remote diamond-based nodes. Unlike previous post-selected operations, the authors achieve a deterministic protocol by integrating high-fidelity remote entanglement, single-shot ^{13}C nuclear spin readout, and real-time feed-forward control. The gate's robustness is verified through the creation of a four-partite GHZ state, confirming genuine entanglement across the remote registers.

The study's primary contribution lies in its real-time phase tracking and feedback, which mitigates dephasing during probabilistic entanglement generation. By enabling complex multi-node operations without post-selection, this work provides a scalable framework for error-corrected distributed quantum computation and the development of a functional quantum internet.

This manuscript addresses a topic of significant contemporary interest within the quantum information science community, and I anticipate it will attract substantial attention from researchers across the field. The authors present a study that is both technically sound and methodologically rigorous; the experimental results are solid and provide convincing evidence for their claims. Given the high quality of the work and the clarity of the presentation, I find the results to be robust and have no further technical concerns or requests for revisions.

We thank the reviewer for their time and effort and their positive evaluation.

However, before the publication, I have questions.

(1) In the implemented real-time phase tracking, for how long can the cumulative tracking error be tolerated? Furthermore, is this precision maintained in cases requiring even longer communication attempts, such as when the distance between nodes is further increased?

For both phase tracking methods, the cumulative error can be contained when the phase is calibrated in "batches", as explained in the Supplementary S3. We calibrate the phase with attempts from 1 to 20 and extrapolate to 50 attempts. To extend this beyond our entanglement attempts timeout (set at 50), we could fit another curve containing the cumulative phase for attempts from 50 to 70 and so on.

For longer entanglement attempts, the extra phase that needs to be added is a deterministic and known contribution and therefore it will in principle not change the error.

(2) I have a particular interest in the scalability of the proposed system. While the current experiment successfully utilizes two NV-center nodes, could the authors comment on how the inherent inhomogeneities between nodes—specifically variations in optical transition wavelengths and hyperfine coupling strengths to nuclear spins—might impact the overall gate performance? Furthermore, do the authors anticipate that such inhomogeneities will present significant challenges when scaling the network to three or more nodes? If so, I would appreciate the authors' insights on potential strategies or compensation mechanisms to mitigate these effects in future large-scale implementations.

As scaling to multiple nodes is not the topic of the current work, we refer the reviewer to our work on a multi-node quantum network and long-range networks [Refs 16,17,18] that discussed these challenges. In short, the variation in optical transition wavelengths is shown to be tunable on all three nodes. Alternatively, the introduction of a quantum frequency conversion system to telecom wavelengths can also compensate for inhomogeneities [see Ref. 16].

The hyperfine coupling strength to nuclear spins is also different from node to node. The qubit registers used here and in previous works have been pre-selected based on these couplings, rejecting about 30% of NV centers. Beyond this pre-selection, a range of hyperfine couplings can be used for high-fidelity control [see Ref. 21]. Isotopic materials engineering may alleviate this need for pre-selection in the future.

Reviewer #2 (Remarks to the Author):

The manuscript reports an impressive experimental advance toward distributed solid-state quantum computing, namely the realization of an unconditional teleported non-local C-NOT gate between remote diamond-based qubit registers, together with the generation of remote four-qubit entanglement. The work is timely, technically strong, and of clear interest to the quantum networking and modular quantum computing communities. I have only a few minor comments, mainly concerning figure clarity and

consistency of notation/presentation, which I believe can further improve the readability of the manuscript.

We thank the reviewer for their time and effort and their positive evaluation, as well as their suggestions for improvement.

Figures 1 and 4a (local operations).

In Figs. 1 and 4a, the assignment/order of the local-operation blocks between Alice and Bob seems visually inconsistent, or at least unclear. Please check whether the upper and lower local-operation parts are swapped, or clarify the control/target convention more explicitly in the figure and/or caption.

The circuit is correct in both figures. The confusion might rise from the fact that we do not implement a direct Controlled-gate with the data qubit as control. Rather, as shown in Fig.4a for Alice, this Controlled-NOT gate is decomposed in NV center native gates.

Figure 2 ($CX_{\pi/2}$ notation).

In Fig. 2, the notation $CX_{\pi/2}$ is not immediately clear. If this is meant to denote an ordinary controlled-X gate in the native decomposition, using simply CX would likely improve readability. Otherwise, the meaning of the $\pi/2$ subscript should be explicitly defined in the figure caption or main text.

We appreciate the suggestion, and we changed the definition in the inset in Figure 2 to CX.

Figure 2a,c (Z_{θ} versus Z_{ϕ}).

Please check the consistency of the phase-angle notation in Fig. 2, in particular Z_{θ} versus Z_{ϕ} . The use of θ may be intentional, but the notation should be made consistent across the manuscript to avoid confusion.

We now explicitly state in the caption that $|\phi\rangle$ represents a generic state on the equatorial plane.

Figure 3a (undefined t).

In Fig. 3a, the symbol t appearing in the expression $\tau-t$ does not seem to be defined in the main text or figure caption. Please define it explicitly in the caption or main text, even if it is explained in the Supplementary Information.

The symbol t was defined in the Methods section, as reported in the caption. We now added a clear definition in the caption.

Figure 3b (vertical axis p_0).

In Fig. 3b, the vertical axis is labeled p_0 , whereas both the caption and the main text appear to describe the plotted quantity as the state fidelity. Please make these consistent and clarify whether the figure shows the $|0\rangle$ population or the state fidelity.

Thanks for spotting the confusing notation. We changed the figures y-label into “Fidelity”.

Figure 5 / main-text description of Alice’s single-qubit gates.

The description of the timing of the single-qubit gates on Alice’s data qubit appears ambiguous. The main text states that these gates are executed “right after the initialization and right after the mid-circuit measurement,” whereas the circuit seems to show more than two such gates, and the final Ygate appears to occur after the feed-forward sequence rather than immediately after the measurement itself. Please clarify the timing and compilation of these gates in the figure and/or text.

Thank you for the question. The gates that belong to the local-operation circuit as highlighted in Figure 1, are represented in purple in Figure 5a. The gates in black represent the phase tracking. We have now clarified this by making it explicit in the caption.

Overall, these are relatively minor issues, and I believe the manuscript will be further strengthened once the above points are clarified. The reported results are significant and will be of broad interest to readers working on quantum networks, distributed quantum processing, and solid-state quantum platforms.

We thank again the Reviewer for the very helpful comments.

Reviewer #3 (Remarks to the Author):

The authors realise a Controlled NOT gate between two remote nuclear spin qubits in two separate experimental setups, a crucial step toward scalable modular quantum processors. They also produced a four-partite GHZ state across two nodes, which is the largest heralded multi-partite entangled state for solid-state systems to date. The ability to operate without post-selection and using real-time feedback enable the realization of this key milestone for distributed quantum computing. The experiment is highly advanced and requires sophisticated protocols and demanding control. Overall, this is a highly relevant and significant work. The main text is somewhat hard to read,

originating from the many advanced techniques involved which are not explained in detail.

I am convinced this is a valuable contribution to Nature Communications, and I recommend publication.

We thank the reviewer for their time and effort and their recommendation.

I only have minor comments:

- Abstract: What should fully-integrated system mean? This doesn't seem to be a particular property of the reported experiment.

Fully-integrated system here refers to the combination of quantum hardware with a dedicated networking software stack. As the abstracts notes, our results demonstrate a key capability that will enable more complex testing of such systems (and indeed not that our results themselves show such a system). We changed fully-integrated system to full-stack system, which we believe better explains our motivation.

- Use of capital letters (RadioFrequency, MicroWave, Dynamical Decoupling) is somewhat unusual

We have now changed these to the more conventional form.

- Power-broadened 575nm pulse – unclear wording – probably the pulse power-broadens the transition?

Indeed. To avoid confusion we changed this to “...using a high-power 575nm pulse”.

- How is DC Stark tuning at high mag field enabled by efficient charge repumping?
Unclear sentence, please rephrase / explain

This has to do with the challenge that at high magnetic field the optical transitions of NV0 show a spin-splitting that is larger than the typical power broadening of the yellow pulse used for lower magnetic field. Therefore, there are two possible options for efficient repumping: using two yellow pulses, each one on resonance with one spin-selective transition, or using a high-power single pulse with frequency parked in the middle of the two transitions. We opted for the second one. This is included in the Supplementary (S1).

- The scheme requires classical communication – how limiting will this be for long-distance links?

For long-distance links, this means that the coherence time of the data qubits needs to exceed the round-trip time of the classical communication between the nodes or between node and midpoint. As shown in Table S3, the data qubits have – even without dynamical decoupling - dephasing times around 10 milliseconds (corresponding to a roundtrip distance of 2000 km in optical fiber) so this is not a relevant bottleneck.

- What limits the fidelity of the data qubit initialization? It is rather low for Alice (85%), and a brief discussion would be helpful in the main text.

85% is the result of imperfect initialization and imperfect assisted readout. For the readout, we are able to quantify its contribution to the infidelity, as included in the supplementary. This implies that the initialization infidelity is rather low.

- What limits the fidelity of the entanglement generation? At least a brief statement would be helpful in the main text.

A detailed investigation of the factors affecting the fidelity in the single-click protocol is included in Ref. 29, as stated in main text. The study in Ref. 29 is conducted on the same setup used in this demonstration.

- What is the entanglement generation rate? Please give a value

- What is the success probability per attempt of successful remote CNOT gates?

The relevant numbers are reported in the supplementary section S6. The data rate is in the range (23-42) mHz. We believe the entanglement generation rate to be of the same order of magnitude, given that the main overhead time is given by the Charge-Resonance Check. The entanglement probability is 1.1×10^{-5} . This probability is now also included in S6.